# Productivity and Nutritional Quality of Nitrogen-Fixing and Non-Fixing Shrub Species for Ruminant Production

**Magnolia Tzec-Gamboa** [1], **Oscar Omar Álvarez-Rivera** [2], **Luis Ramírez y Avilés** [1], **Juan Ku-Vera** [1] and **Francisco Javier Solorio-Sánchez** [1,*]

[1] Faculty of Medicine Veterinary and Animal Sciences, University of Yucatán, Km 15.5 Carretera Mérida-Xmatkuil s/n, Mérida 97315, Mexico; magnolia.tzec@correo.uady.mx (M.T.-G.); luis.ramirez@correo.uady.mx (L.R.y.A.); kvera@correo.uady.mx (J.K.-V.)

[2] Biotechnology Unit, Scientific Research Center of Yucatan A.C., Calle 43 No. 130 by 32 and 34, Col. Chuburna de Hidalgo, Mérida 97205, Mexico; oscaralvarez.uady@gmail.com

\* Correspondence: ssolorio@correo.uady.mx

**Abstract:** Fodder shrubs are important dry season feed sources for livestock in semi-arid environments. It has been proposed that a mixture of leguminous with non-leguminous shrubs may increase rates of N cycling and improve biomass and fodder quality. The objective of the present study was to assess the biomass productivity and fodder quality of leguminous shrubs growing a mixture with non-leguminous shrubs. Three shrub species—the legume *Leucaena leucocephala* (Lam.) and the non-legumes *Guazuma ulmifolia* (Lam.) and *Moringa oleifera* (Lam.)—were grown as legume/non-legume mixtures and as monocultures. Total fodder production and quality were estimated over five harvests at three-month intervals. The *Leucaena–Guazuma* mixture had the largest fodder production (9045 kg ha$^{-1}$ year$^{-1}$), followed by the *Leucaena* monocrop (7750 kg ha$^{-1}$ year$^{-1}$). Total nitrogen accumulation in the foliage was also high in the *Leucaena–Guazuma* mixture, reaching 282 kg ha$^{-1}$ year$^{-1}$; the *Leucaena* monocrop accumulated 244 kg N, while the *Moringa* monocrop took up only 46 kg N ha$^{-1}$. The concentration of polyphenols was high in *Leucaena* and *Guazuma*, while *Moringa* had the lowest value. The high survival and excellent growth rates, as well as the high foliage production observed in *Leucaena* and *Guazuma*, suggest that they have the potential to provide high-quality fodder for livestock.

**Keywords:** alley systems; fodder production; chemical content; tannins

## 1. Introduction

Tropical regions are characterized by an abundant supply of solar energy and a great diversity of natural resources, which give them a high potential for food production. However, the tropics are also more sensitive to natural disturbances such as droughts or excessive rainfall. The dynamics of this type of event makes tropical regions complex, highly variable and generally low in agricultural productivity. In addition, these regions are still experiencing rapid population growth. The consequences commonly involve a further deterioration in already degraded land resources and the migration of farmers in search of better agricultural land. These factors increase the need to look for reliable and sustainable methods of food and feed production. In the Yucatan peninsula, farmers have retained the ancient traditional shifting agriculture system (Milpa) from Mayan culture [1,2]. The ancient Maya probably selected many of the leguminous nitrogen-fixing species, leading to the great preponderance of this botanical family (Fabaceae) in the Yucatan peninsula [3,4]. However, although the Mayan farmers practiced mixed farming with crops and animals for centuries, currently, these agricultural systems are not integrated. Animal production, particularly of cattle, is based mainly on seasonal grass production. Generally, pasture lands based on mono-crop-grass are characterized by their high productivity (in terms of dry matter biomass) during wet periods; in contrast, grass forage is scarce and/or of poor

quality during dry periods [5], thus directly affecting livestock productivity. To address the issue of feed shortage, many developing countries import significant quantities of grain and processed feed from developed nations. This dependence of developing countries on imported agricultural products can make farming systems unsustainable in the long term, not only due to the dependence on importations but also through system outputs due to the loss of nutrients from the soil.

The use of trees and shrubs to supplement fodder supplies has enormous potential for addressing some of the problems associated with livestock feed constraints (quantity and quality) in tropical regions. In general, tropical trees and shrubs produce a considerable amount of foliage that is high in nutrients—particularly nitrogen, which is one of the main constraints on tropical grassland productivity—throughout the entire year. Moreover, leguminous trees can fix nitrogen throughout the year and also absorb nutrients from deeper soil horizons, which are also relocated to the surface through litter decomposition [6,7]. One way of taking advantage of this process is to grow nitrogen-fixing trees in pastures [8]. Spatial integration of animals with shrub forage production can limit the productivity of the system, as animals could damage the shrubs, compromising the sustainability of the system. This study investigates the possibility of growing different shrub species in fodder bank systems (where they cut and carry the biomass to feed animals). As nitrogen fixation results in a significant energy cost to the plant, the growth rate of non-leguminous trees is expected to exceed that of leguminous trees in situations where nitrogen is non-limiting. In nitrogen-limiting sites, mixed stands of leguminous and non-leguminous species present an opportunity for optimizing total stand growth and quality [9–11]. Examples of such forage shrub species are *Leucaena leucocephala*, *Guazuma ulmifolia* and *Moringa oleifera*. These species are frequently grown in southeast Mexico and are characterized by rapid growth and biomass production, deep rooting, ease of establishment, tolerance of repeated coppicing and resprouting ability, foliage palatability, wide adaptability and stress tolerance [12,13].

The potential benefits of such mixed systems for feeding livestock have been well documented [14–17]. However, rather little is known about the agronomic aspects of such systems, particularly the relationship between different species growing in the mixture. Most of the research on woody plants in fodder banks has focused on monoculture systems, while other studies have focused on the inclusion of multiple species but in rotational systems more than in mixed systems. Consequently, the extent to which one species enhances the biological performance of another is not well known. The objectives of the present study are as follows: (1) to evaluate the biomass yield and fodder quality of monoculture and mixed fodder banks of leguminous (*Leucaena leucocephala*) and non-leguminous (*Guazuma ulmifolia* and *Moringa oleifera*) woody species; (2) to assess the main factors affecting biomass yield and quality of fodder bank species in mixtures. These shrub species have been well accepted among farmers in the region, and it has been observed that they have good growth under limiting soil and weather conditions of the region. Establishing them in mixed fodder banks is expected to enhance their beneficial interactions, which should be positively reflected in biomass production and forage quality.

## 2. Materials and Methods

### 2.1. Site Description

The study was carried out at the research field station of the Faculty of Medical, Veterinary and Animal Sciences, University of Yucatán, Mérida, in the Yucatán peninsula, southern Mexico. The experimental area lies at 21°15′ N and 90°25′ W at an altitude of 10 m a.s.l. in the sub-humid climatic zone, with an average annual rainfall (highly variable) of 960 mm and a 6–7 month dry period [18]. The annual average temperature is 26 °C, ranging from a daily average of 23 °C (max 32 °C, min 15 °C) from November to February to 30 °C (max 37 °C, min 23 °C) from March to September. The landscape is an undulating karstic plain, and the soils are calcareous and mainly shallow (<0.30 m depth), with much of the surface being exposed to limestone outcrops or loose rocks [19] and some areas of low

forest vegetation. Dominant soils are classified as Leptosols and are of moderate fertility, with 1.0 to 1.5% organic carbon content and a pH range of 7.5 to 7.8.

## 2.2. Nursery Management and Establishment

One nitrogen-fixing species, *Leucaena leucocephala* (Lam.) de Wit, and two non-nitrogen-fixing species, *Guazuma ulmifolia* Lam. and *Moringa oleifera* Lam., were selected for the experiment. Shrub containers were made from $100 \times 150$ mm of local commercial polythene black bags with pleated bottoms, and with perforations in the lower part of the bag to allow good water drainage. Bags were filled with local soil (2 mm sieved) sourced close to the experimental site. Once the seeds germinated, the seedlings were irrigated every second day and kept free of weeds by manual control. The *Leucaena* and *Guazuma* seeds were treated by soaking them for 30 seconds in water at 90 °C to break the seed coat. The seeds of each of the species were deposited in individual bags. After 6 weeks, the seedlings were transplanted (15 July 2012) into east–west oriented rows to minimize light competition between the trees.

## 2.3. Experimental Design

The experiment consisted of five treatments each replicated four times (*N*= 20 plots), within a 0.5 ha experimental field:

1. 50:50 mixture of *Leucaena* and *Guazuma*;
2. 50:50 mixture of *Leucaena* and *Moringa;*
3. *Leucaena* monoculture;
4. *Guazuma* monoculture;
5. *Moringa* monoculture.

These five treatments were arranged in a randomized block design, with five tree rows per block. Blocks measured $10 \times 20$ m, with a 2 m space between rows and with a 0.5 m tree spacing within rows. The 50:50 mixtures had one seedling of each species of the combination planted in each planting hole, while the monoculture treatment had only one seedling in each hole. The experiment lasted for 26 months (July 2012–August 2014); measurement dates are provided in months after transplanting. Shrubs were first pruned by removing all biomass above 60 cm height 42 weeks after transplanting (end of July). All regrowth above 60 cm was harvested three months after the end of the rainy season (end of November) and thereafter at three-month intervals.

## 2.4. Chemical Analysis

Soil was sampled shortly before the seedlings were transplanted (baseline); five composite random samples were taken from each of the twenty plots, from 0 to 10 cm depth, after removing the litter layer. Soil samples were air-dried, sieved to a 2 mm particle size and analyzed for a $pH_{H2O}$ ratio of 1:1 [20]. Total N was measured using the Kjeldahl technique [21] and total organic carbon was determined by Walkley–Black method [22]. Available P (Olsen, for calcareous soils) was measured colorimetrically after extraction with $NaHCO_3$ [23]. The exchangeable Ca, Mg and K were extracted with 1 M ammonium acetate solution for calcareous soils and determined by atomic absorption spectrophotometry [24]. Additionally, the stone content of the soil was determined in the experimental area [25].

On the other hand, for plant material, after 11 months of growth, the prunings were separated into leaves, edible stems (<5 mm diameter) and branches (>5 mm diameter). Fresh weight of all biomass fractions was obtained in the field before taking sub-samples for dry weight determination (oven drying at 60 °C for leaves and edible stems, and at 70 °C for woody tissue, both to constant weight). Once dried, the samples were combined by plot and biomass component, ground to pass through a 1.0 mm screen and homogenized for chemical analysis.

Total N was determined by the Kjeldahl method [19]. Neutral detergent fiber (NDF) and acid detergent fiber (ADF) were determined following the methods proposed by [26]. Total extractable polyphenols were determined colorimetrically using the Folin–Ciocalteu method, while

total and condensed tannins (proanthocyanidins) were determined by the polyvinylpolypyrrolidone (PVPP) and the butanol–HCl methods, respectively [27,28]. Subsequent harvests were separated into leaves, leaves + edible stems (the "fodder" fraction) and woody fractions. Only the leaf and fodder components were analyzed as above. The chemical data for the edible stem component were then calculated from the results for these two fractions.

### 2.5. Tree Measurements

Species survival rates were calculated for 1, 11 and 24 months after transplanting; mean stem diameter and height were monitored throughout the study period for each species in all treatments. Tree measurements were taken at 1, 3, 6 and 11 months. On each occasion, the height (cm) and diameter (mm) 5 cm above ground level in the three central rows of each plot were measured using a 3.00 m measuring stick and a 210 mm caliper, respectively, and the plot means were calculated. The leaf/edible stem dry matter ratio was measured for four branches from each replicate per treatment, selected at random.

Notation: The following notations for the species in mixtures are used in the text, tables and figures below: *Leucaena* (G) indicates *Leucaena* component growing in a mixture with *Guazuma*, while *Guazuma* (L) refers to the *Guazuma* component in the same mixture. Similarly, *Leucaena* (M) indicates *Leucaena* growing in a mixture with *Moringa*, while *Moringa* (L) refers to the *Moringa* component in the same mixture. Additionally, fodder is defined (as indicated in the previous section) as the edible biomass (sum of foliage and stems thinner than 5 mm diameter) obtained in each pruning.

### 2.6. Statistical Analysis

The statistical analysis was carried out using SPSS version 21.0 (SPSS Inc., Chicago, IL, USA). All data were tested to fit the assumption of normality using a Kolmogorov–Smirnov test and the homogeneity of variances with Levene's test ($p < 0.05$), and non-normal data were transformed as necessary. The analyses of variance (ANOVAs) were calculated on plot means with log-transformed data. To test for significance in ANOVAs, means were compared using Tukey's test.

## 3. Results

### 3.1. Environmental Variables

Weather conditions during the 26 months of the experiment are shown in Figure 1. September of the first year had a remarkably high rainfall compared with all other months of the study, primarily due to the occurrence of a hurricane, while January and March were completely dry. Mean maximum and minimum temperatures for the period were 27 to 39 °C and 12 to 25 °C, respectively. May was the hottest month (39 °C) and December was the coldest month (12 °C).

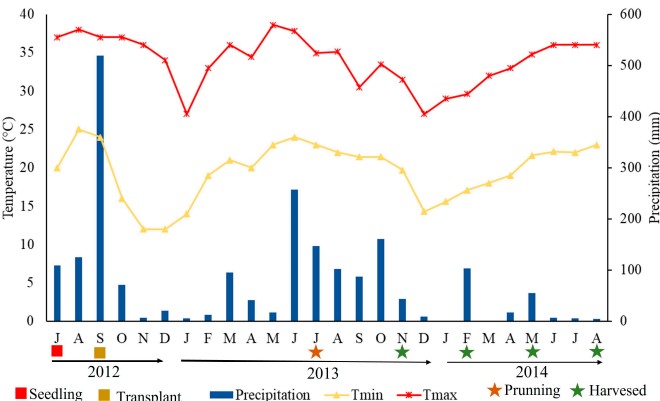

**Figure 1.** Weather conditions during the first 26 months after transplanting the seedlings.

### 3.2. Soil Chemical and Physical Characteristics

Soil characteristics of the experimental site are shown in Table 1. Blocks III and IV showed substantially higher P contents (mg kg$^{-1}$) than the other two blocks. In general, all blocks had high stone contents (average 74% by weight, 55–60% by volume) in the top 10 cm of soil. Soil pH was very similar in all blocks, with a mean of 7.8.

**Table 1.** Description of some physical and chemical properties of the soil for the experimental area.

| Block | Stone | pH | N | C | P | Exch K | Exch Ca | Exch Mg |
| | (%) | | % | | | Mg kg$^{-1}$ | | |
|---|---|---|---|---|---|---|---|---|
| I | 78 | 7.8 | 0.89 | 6.4 | 28 | 530 | 872 | 352 |
| II | 60 | 7.8 | 0.98 | 5.0 | 45 | 565 | 824 | 328 |
| III | 79 | 7.9 | 0.99 | 7.2 | 81 | 457 | 1077 | 310 |
| IV | 79 | 7.9 | 0.96 | 6.1 | 111 | 517 | 1573 | 388 |
| Mean | 74 | 7.8 | 0.95 | 6.2 | 66 | 517 | 1086 | 345 |

### 3.3. Tree Survival in Mixtures and Pure Stands

More than 85% survival was achieved during the first year for all the species. However, mean survival rates differed significantly ($p < 0.05$) at 1 and 11 months (Table 2). The mixtures formed by *Leucaena* (*Guazuma*) and *Leucaena* (*Moringa*), and the *Guazuma* monoculture, all achieved more than 95% survival rates. However, after 24 months, the treatment means were neither agronomically nor statistically significantly different (Table 2).

**Table 2.** Tree survival (%) at 1, 11 and 24 months after transplanting seedling into the field.

| | Survival (%) | | |
| Treatment | 1 Month | 11 Months | 24 Months |
| | Mean | Mean | Mean |
|---|---|---|---|
| *Moringa* | 91 a | 85 a | 69 a |
| *Moringa* (L) | 93 ab | 90 ab | 76 a |
| *Guazuma* | 97 ab | 95 ab | 88 a |
| *Guazuma* (L) | 97 ab | 96 ab | 80 a |
| *Leucaena* | 97 ab | 89 ab | 80 a |
| *Leucaena* (M) | 98 ab | 97 b | 81 a |
| *Leucaena* (G) | 100 b | 98 b | 86 a |
| SED | 2.5 | 3.4 | 6.8 |

SED: Standard error of the differences between treatment means; means in a column followed by the same letter are not significantly different using Tukey's test at $p = 0.05$.

### 3.4. Shrub Growth in Pure and Mixed Stands

Height and diameter are shown in Table 3, after species had been growing for a six-month period. Growth of all species in all treatments was fast after the first three months and faster after six months. Estimated mean tree heights differed significantly between treatments ($p < 0.05$) at one and three months after planting (Table 3). Heights ranged from 0.17 to 0.43 m for *Guazuma* and *Leucaena* monocultures after one month. However, *Guazuma* trees (in mixture and in monoculture) were the shortest, and there were no statistical differences when compared with the other single species and mixtures at the last two sampling points. Plant height measurements also revealed that *Guazuma* and *Moringa* generally were not affected by mixing them with *Leucaena*. Eleven months after planting, *Leucaena* and *Leucaena* (G) had grown faster, having reached 2.22 and 2.00 m, respectively, while *Guazuma* had grown slower and tended to be shorter than the other species (Table 3).

**Table 3.** Tree heights (m) and diameters (mm) during the study period.

| Species | Months after Planting | | | | | | | |
|---|---|---|---|---|---|---|---|---|
| | **1** | | **3** | | **6** | | **11** | |
| | **Height** | **Diameter** | **Height** | **Diameter** | **Height** | **Diameter** | **Height** | **Diameter** |
| *Moringa* | 36 b | 6 b | 38 ab | 9 bc | 84 | 18 | 180 | 29 |
| *Moringa* (L) | 31 b | 6 b | 38 ab | 9 bc | 83 | 17 | 193 | 28 |
| *Guazuma* | 17 a | 4 a | 22 a | 6 ab | 54 | 14 | 119 | 24 |
| *Guazuma* (L) | 18 a | 4 a | 26 ab | 6 ab | 62 | 13 | 131 | 21 |
| *Leucaena* | 43 b | 4 a | 47 b | 6 ab | 84 | 16 | 222 | 29 |
| *Leucaena* (M) | 32 b | 4 a | 33 ab | 5 a | 60 | 12 | 178 | 23 |
| *Leucaena* (G) | 35 b | 4 a | 41 ab | 6 ab | 79 | 14 | 201 | 24 |
| SED | 3.71 | 0.53 | 6.56 | 1.11 | 17.6 $^{ns}$ | 2.59 $^{ns}$ | 45.7 $^{ns}$ | 4.20 $^{ns}$ |

SED: Standard error of the differences between treatment means; means in a column followed by the same letter are not significantly different using Tukey's test at $p < 0.05$. $^{ns}$ = no significance.

Tree stem diameters in all treatments (single species or mixed stands) increased with time (Table 3). Mean stem diameter growth rate was highest in the three-to-six-month period, both in single and in mixed-species stands, although the rate of increase in the cross-sectional area continued to increase. Treatment means were significantly different ($p < 0.05$) at one and three months. *Leucaena* and *Moringa* growing in monoculture generally had a faster increase in diameter than monoculture *Guazuma* (Table 3). The rate of diameter increase was not adversely affected for any species when mixed with other species.

There were no significant differences in stem diameter at 11 months with the mean diameter ranging from 21 mm for *Guazuma* (L) to 29 mm for *Moringa* and *Leucaena* in monoculture (Table 3). Although there were no statistically significant differences between treatments at 11 months, qualitative observations suggested that *Guazuma* tended to be the shortest species, with a flattened leaf canopy and numerous thin stems; meanwhile, *Leucaena* plants in monoculture were mostly multi-stemmed and densely packed with long branches and, compared to *Moringa*, presented a denser aboveground canopy in the first period of growth.

### 3.5. Biomass Yield

Fodder production in each of the five harvests is shown in Table 4 along with the respective cumulative totals. Fodder production after 11 months showed large variations between the treatments ($p < 0.05$). The *L. leucocephala* monocrop and the *Leucaena–Moringa* mixture showed the highest production at the first pruning. The *Guazuma* monocrop and *Leucaena* (M) also showed significantly greater fodder production than the *Moringa* monocrop (* $p < 0.05$).

**Table 4.** Estimated cumulative fodder production (kg DM/ha$^{-1}$) from mixed and pure tree mixture stands.

| Species | July | November | February | May | August | Total ⌐ |
|---|---|---|---|---|---|---|
| | **1** | **2 \*** | **3 \*** | **4 \*** | **5 \*** | **(kg ha$^{-1}$)** |
| *Moringa* | 510 a | 440 a | 258 a | 457 a | 265 a | 1930 |
| *Guazuma* | 1167 ab | 1384 b | 259 a | 1332 a | 952 b | 5094 |
| *Leucaena* | 2456 bc | 1698 b | 1171 b | 1413 a | 1012 b | 7750 |
| *Leucaena* (M) | 1613 ab | 1329 b | 925 b | 1086 a | 918 b | 5871 |
| *Leucaena* (G) | 2911 bc | 1714 b | 1027 b | 1894 a | 1499 b | 9045 |

July, November, February, May, August = Fodder accumulated in the three months interval between prunings. ⌐ Total = sum of the first harvest plus the last four harvests. * = Harvest numbers. Different lowercase letters indicate significant differences.

In November, after three months of regrowth, significant differences ($p < 0.01$) in fodder production among the treatments were found. Production ranged from 440 to 1714 kg DM/ha$^{-1}$. *Moringa* in monoculture had the lowest yield, while the highest production was observed in the *Leucaena–Guazuma* mixture, followed by *Leucaena* in monoculture with 1698 kg DM/ha$^{-1}$ (4154 kg DM/ha$^{-1}$ accumulated in the two first harvests). In February, *Leucaena* had the highest production of the monocultures with 1171 kg DM/ha$^{-1}$ (5325 kg DM/ha$^{-1}$ accumulated in three harvests). A significant difference ($p < 0.01$) was observed in the *Leucaena–Moringa* and *Leucaena–Guazuma* mixtures with respect to the non-fixing trees in the monoculture stands, which had the lowest fodder production. Among mixtures, *Leucaena* (M) gave the least fodder production with 925 kg DM/ ha$^{-1}$ in the third (February) harvest and 3867 kg DM/ha$^{-1}$ accumulated over the first three harvests (July, November and February).

In May, fodder production was highest in *Leucaena* (G), with 1894 kg DM/ha$^{-1}$ (6134 kg DM/ha$^{-1}$ accumulated), compared with the other mixtures. Mixing *Leucaena* with *Guazuma* increased the fodder production of *Leucaena*, resulting in 480 kg DM/ha$^{-1}$ more *Leucaena* fodder in mixture than *Leucaena* in monoculture. In the same way, mixing *Guazuma* with *Leucaena* increased its fodder production, resulting in 560 kg DM/ha$^{-1}$ more fodder for *Guazuma* in mixture than that produced by *Guazuma* in monoculture. In the last pruning (August), *Moringa* in monoculture showed the lowest fodder production ($p < 0.01$), with only 265 kg DM/ha$^{-1}$ (total production 1420 kg DM/ha$^{-1}$). Mixing species generally resulted in increased foliage yield, particularly in *Leucaena–Guazuma*, which had a total production of 6134 kg DM/ha$^{-1}$. *Leucaena* on its own or *Leucaena* (M) had intermediate fodder production with 5294 and 4258 kg DM/ha$^{-1}$, respectively.

Figure 2 illustrates the estimated cumulative (last four prunings) production of leaves, edible stems and total fodder for the species in mixture and single-species stands. Annual leaf production (measured from 1 to 2 years after planting) ranged from 964 to 3750 kg DM/ha$^{-1}$, equivalent to 68–61% of net fodder production. Estimated total fodder production from the *Leucaena–Guazuma* mixture greatly exceeded that of the other species ($p < 0.01$). At an age of 2 years, the *Leucaena* monocrop had yielded significantly ($p < 0.01$) more fodder than the other species in monoculture, whereas *Leucaena–Guazuma* produced 36% more fodder than the *Leucaena–Moringa* mixture.

Estimated total fodder production by *Leucaena–Guazuma* mixture was always higher compered with the other species, except for the second pruning (February), when the *Leucaena* monocrop had a greater production. In contrast, the *Moringa* monocrop always had the lowest production, below 500 kg ha$^{-1}$ per pruning (Table 5). It is interesting to note that most of the total biomass yield was in the form of leaves and edible stems, with very little woody material recorded in the first harnessing (after four months of the first pruning). It is also important to mention that 60–68% of the fodder biomass was made up of leaves (Figure 2). Among the mixed treatments, *Leucaena* (G) had the largest production (6133 kg ha$^{-1}$ year$^{-1}$) with 60% in the form of leaves, and the mixture formed by *Leucaena–Moringa* showed an intermediate production (3926 kg ha$^{-1}$ year$^{-1}$) with 61% in the form of leaves (Figure 2).

*3.6. Chemical Composition of Pruned Foliage at Three-Month Regrowth Intervals*

Table 5 presents the chemical composition (average of five prunings) of the different fractions obtained from the treatments. As expected, N in the legume leaves (3.6–3.7%) was higher than in the non-leguminous species. The *Guazuma* monocrop had the lowest content (2.0–2.4%). A similar tendency was observed in the fodder; *Leucaena*, whether in monocrop or in mixture, had the highest N content (3.1%) while *Guazuma* monocrop had the lowest (1.8–1.9%). Meanwhile, *Moringa* had intermediate leaf N content values, ranging from 2.6 in mixture with *Leucaena* to 3.0% in monocrop. The fodder fraction of *Moringa* as a monocrop and *Moringa* (L) had almost identical N contents (2.4–2.5%).

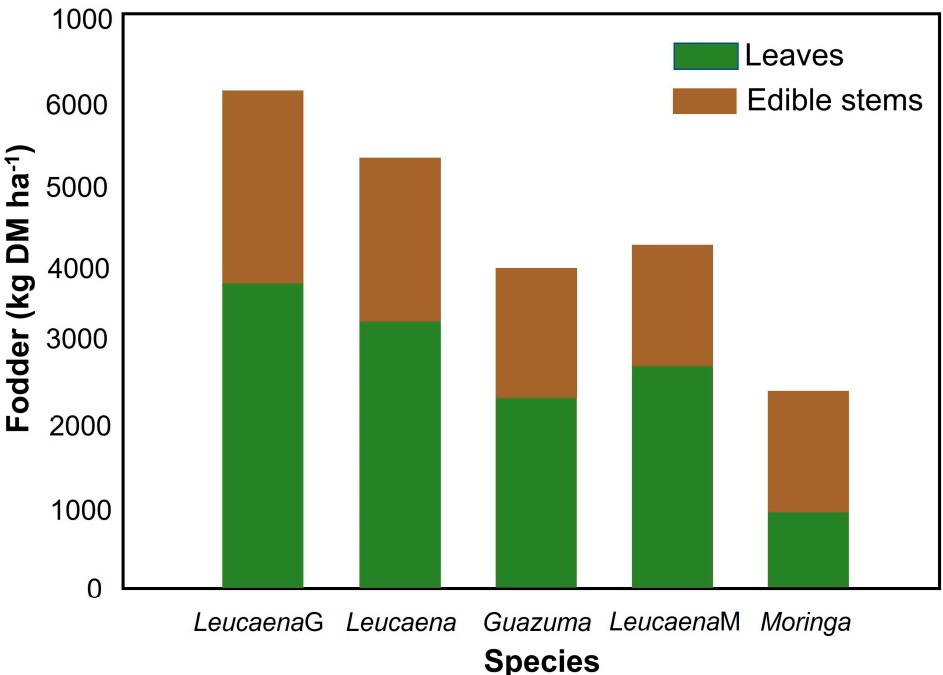

**Figure 2.** Estimated cumulative (mean) fodder production from shrubs growing as monocrops in mixed stands.

**Table 5.** Chemical characteristics of different plant components (on dry matter basis).

| Component | Content in Prunings (%) | | | | | | |
|---|---|---|---|---|---|---|---|
| | N | CP | NDF | ADF | Pp | Tt | Ct |
| | | | | Leaves | | | |
| *Leucaena* | 3.6 | 23 | 36 | 21 | 5.3 | 4.2 | 2.7 |
| *Leucaena* (M) | 3.7 | 23 | 36 | 22 | 5.1 | 4.1 | 2.7 |
| *Leucaena* (G) | 3.5 | 22 | 35 | 21 | 5.2 | 4.3 | 2.3 |
| *Moringa* | 3.0 | 18 | 31 | 21 | 3.2 | 2.6 | 0.5 |
| *Moringa* (L) | 2.6 | 16 | 33 | 23 | 2.9 | 2.2 | 0.5 |
| *Guazuma* | 2.0 | 12 | 37 | 21 | 4.3 | 3.6 | 3.5 |
| *Guazuma* (L) | 2.4 | 15 | 36 | 21 | 4.0 | 3.4 | 3.1 |
| | | | | Twigs | | | |
| *Leucaena* | 0.5 | 3.5 | 40 | 24 | 1.2 | 0.95 | 0.33 |
| *Leucaena* (M) | 0.6 | 4.0 | 40 | 25 | 0.2 | 0.18 | 0.05 |
| *Leucaena* (G) | 0.4 | 2.5 | 39 | 24 | 0.9 | 0.95 | 0.38 |
| *Moringa* | 0.6 | 3.8 | 37 | 26 | 0.8 | 0.70 | 0.00 |
| *Moringa* (L) | 0.2 | 0.8 | 34 | 25 | 0.7 | 0.60 | 0.05 |
| *Guazuma* | 0.2 | 1.8 | 39 | 24 | 0.2 | 0.07 | 0.15 |
| *Guazuma* (L) | 0.5 | 2.8 | 39 | 24 | 0.3 | 0.15 | 0.13 |
| | | | | Fodder * | | | |
| *Leucaena* | 3.1 | 20 | 46 | 30 | 4.2 | 3.2 | 2.3 |
| *Leucaena* (M) | 3.1 | 20 | 45 | 30 | 4.9 | 3.9 | 2.7 |
| *Leucaena* (G) | 3.1 | 20 | 45 | 30 | 4.3 | 3.3 | 1.9 |
| *Moringa* | 2.4 | 15 | 41 | 29 | 2.5 | 1.9 | 0.5 |
| *Moringa* (L) | 2.5 | 16 | 40 | 30 | 2.2 | 1.6 | 0.5 |
| *Guazuma* | 1.8 | 11 | 47 | 31 | 4.1 | 3.5 | 3.7 |
| *Guazuma* (L) | 1.9 | 12 | 46 | 31 | 3.8 | 3.2 | 3.0 |

CP = crude protein; NDF = fiber content; ADF = acid detergent fiber; Pp = polyphenol; Tt = total tannins; Ct = condensed tannins; * weighted mean of leaves + twigs.

Fiber content (NDF) differed considerably between species. *Moringa* leaves had the lowest content (31%) compared with the *Guazuma* monocrops (37%). A similar trend was observed in the fodder component, where *Moringa* had 40% while *Guazuma* fodder had the highest concentration (47%). In contrast, acid detergent fiber (ADF) differences between treatments were relatively small, with values ranging from 21 to 23% for leaves and from 29 to 31% for fodder (Table 5). Most of the *Leucaena* and *Guazuma* leaf and fodder samples had high contents of polyphenols and total tannins. On the other hand, *Moringa* leaves and fodder components (from monocrop or from mixtures) had negligible or undetectable amounts of tannins compared with *Guazuma* (3.5–3.7%).

## 4. Discussion

### 4.1. Tree Survival and Growth

By the end of the experimental period, survival rates did not show large differences between the treatments. In areas where there is a known risk of low survival, planting density can be increased to compensate. Although the results presented here refer only to the two years after planting, it is believed that trees that had survived thus far were likely to persist unless competition for nutrients or water becomes severe or serious damage is caused by pests or diseases. Although pruning limits the size of the individual tree canopy, the stem and below-ground components of the tree continue to grow, and nutrients are removed from the system when the foliage is used as fodder. As the demand for P of N-fixing species is high, while the trees are getting older, additional P is needed to prevent this element limiting these trees' productivity, particularly where there is competition from other trees [29,30].

Tree survival and growth rates for *Leucaena* in monoculture and in mixed stands in this study were within the range of values reported for this species in Puerto Rico by [10], where its performance was studied in mixtures with *Casuarina equisetifolia* and *Eucalyptus robusta*. The performance of *Leucaena* was unaffected by mixing it with *Guazuma*. Mean survival rates were also remarkably similar for the *Leucaena* monocrop and for *Leucaena* mixed with *Moringa*.

The results of the present experiment show that, during the first few months after planting, *Moringa* grew rapidly enough to compete effectively with *Leucaena*, with the mean stem diameter of the former being almost the same in the mixture as in the *Moringa* monoculture, whereas the diameter of *Leucaena* was lower in the mixture (Table 3). In contrast, *Guazuma* diameter growth tended to be smaller in the mixture than in the monoculture, with significant differences ($p < 0.05$) only for the two first assessments.

Furthermore, *Guazuma* grew with a small bush-like habit, different in structure from the other species of this study, and tended to produce large branches relative to its stem size. Such differences in plant morphology between *Leucaena* and *Guazuma* are thought to be an advantage in mixtures.

Binkley et al. [31] noted that species' growing in mixtures are expected to capture solar energy more efficiently due to more effective canopy architecture. The differences in architecture shown by these two species may have contributed to the increased biomass production (Table 4).

### 4.2. Biomass Yield

The results obtained from this work showed significant differences between the mixed stands in the amounts of fodder that they produced: *Leucaena–Guazuma* mixtures produced up to 35% more fodder than the mixture formed by *Leucaena–Moringa*. Among the monocultures, *Leucaena* produced significantly more foliage than did *Moringa* or *Guazuma*. Yields of fodder ranged from 1.9 to 9.0 tons DM ha$^{-1}$. The mixed stand of *Leucaena–Guazuma* produced the greatest biomass, while *Moringa* on its own recorded the lowest. Results for biomass production from the literature include values far below those found in this study. For example, for a three-year period using mixtures of multipurpose trees, mean foliage yields of 100 to 300 kg DM ha$^{-1}$ for *Leucaena* in monoculture and *Leucaena–Sesbania*,

respectively [32]. On the other hand, after a three-year experiment with leguminous and non-leguminous tree species, a mean annual leaf and twig (edible stem) biomass total yield of 3.5 t ha$^{-1}$ was found for *Leucaena* [33].

Fodder production from the single species stands of *Guazuma*, although less than *Leucaena*, exceeded the productivity of *Moringa*. These results indicate the adaptation to local environmental conditions of the local species in comparison to *Moringa*. However, in another study carried out by [34], *Moringa* gave higher biomass and CP than *Leucaena*. These results stress the importance of local environmental conditions and its effect on plant development and production.

Moreover, in this study, *Moringa* performs less well when grown with *Leucaena* than with *Guazuma*. It may be less shade tolerant than *Guazuma* since it does not maintain an adequate amount of foliage when grown in a mixture with *Leucaena*, *Leucaena* may simply develop its canopy more quickly or it may not compete well below-ground. However, the excellent performance of both *Leucaena* and *Guazuma* during the study was not surprising, as it has been observed that they both grow well even under adverse soil and weather conditions.

The low rate of fodder production attained with *Moringa* indicates that it was less well adapted than the other species to the environmental conditions. In addition, it was very susceptible to attack by ants; such an attack on the young shoots a few days after pruning impeded complete recovery of the trees. The results obtained here suggest that *Guazuma* is a shade-tolerant species that can be mixed with another, taller, species such as *Leucaena* without its performance being adversely affected. A similar capacity was reported for *Acacia mearnsii* by [30] in a growth study with a mixed-species plantation.

It is important to consider that the frequency of pruning can affect the productive behavior of forage species. It has been reported that in the case of other legumes such as *Leucaena*, *Gliricidia sepium*, *Sesbania grandiflora* and *Sesbania sesban*, the frequency of pruning can decrease their biomass production [35,36].

*4.3. Nutritional Quality*

The nitrogen accumulation in the foliage of the *Leucaena–Guazuma* mixture (282 kg N) and *Leucaena* alone (244 kg N) was markedly greater than in the foliage of the other shrub species (Table 5). Although the foliage from *Moringa* was higher in N (2.4%) than the foliage from *Guazuma*, because of its greater growth over the experimental period, the *Guazuma* stand accumulated 50% more N than the *Moringa* stand.

The data on nutrient concentrations for *Leucaena* (3.6% N) are similar to those reported by [8] in a study carried out in Valle Nacional in the humid tropics of Oaxaca, Mexico. In terms of their capacity to accumulate N, *Guazuma* and *Moringa* in combination with *Leucaena* can be considered as good multipurpose species for fodder production. Nevertheless, it must be stressed that it is not only the nitrogen content of the foliage that is important for animal food production and nutrition; the content of other plant compounds such as polyphenols, lignin and tannins has been cited as a better parameter for characterizing feed quality and capacity to release nutrients.

Crude protein values found in this study were similar to those reported by [37], who reported a content of 21.9% (DM) in *Leucaena* leaves; in contrast, in a study realized by [35], *Moringa* was the species with the highest crude protein content (31%).

The polyphenol and tannin content found in this study for *Leucaena* and *Guazuma* species were clearly higher than those reported by [38] for the same tropical tree species. They found very low or non-detectable concentrations of phenols and tannin in the leaves of *Leucaena* and *Guazuma*. The differences in these plant compounds may have been caused by the different methodologies used, as both studies were carried out in very similar weather conditions and using the same shrub species as in the present study. Additionally, as the feeding value of *Moringa* and *Guazuma* foliage for livestock has not been intensively investigated [39–41], the lack of data on these species makes it difficult to come to more definite conclusions.

In hedgerow intercropping, shrubs provide a supply of fodder but also recycle nutrients and provide organic matter to the soil. Data compiled by [42] show that leguminous trees in hedgerow intercrops typically produced up to 20 t ha$^{-1}$ year$^{-1}$ of prunings, containing 358 kg of N, 28 kg of P, 232 kg of K, 144 kg of Ca and 60 kg of Mg. Ref. [35] recommended *Leucaena* as a suitable hedgerow species, mainly for the humid and sub-humid tropics. Ref. [43] reported that the prunings of *Leucaena* can provide large and important amounts of nutrients. If the prunings are used for feed, they can correspondingly provide a valuable supply to the ruminant livestock. The establishment of productive systems with multiple arrays allows ruminants to have a diverse diet, which is reflected positively in their nutrition and health [44]. Data are still needed on the non-N nutrient supply to soil or animals from the two non-legume species used in this study. Aspects to consider that may affect the potential for biomass production are the tolerance of the tree species to regular pruning and to pests and diseases; in the same way, the maturity of the leaves influences the quality of the forage, for example, in *Leucaena* it has been reported that the optimum age for cutting is 70 days, but this figure may be affected depending on the environmental conditions of the region [45].

## 5. Conclusions

The fodder banks were established without difficulty and with a high survival rate, assisted by the large amount of rainfall observed during the establishment phase. The biomass production of the species tested in this study, particularly *Leucaena* and *Guazuma*, was sufficient to make them excellent candidates for fodder banks in the environment of Yucatan, both in monoculture and mixed stands. Although *Moringa* grew rapidly in the period before pruning, it appeared to be more sensitive to drought and was the only species of the three to be attacked by pests. The other two species, which are native to Yucatan, are superior in this regard.

The high contents of nitrogen and other nutrients in the foliage of *Guazuma* and *Moringa*, in a mixture with *Leucaena*, make these species a suitable fodder alternative as a food supplement for the livestock; this is especially so during the dry periods when all the sources of grass fodder are scarce and of low nutritive quality. However, other aspects of foliage quality were less desirable. However, too frequent removal of the foliage from the trees will result in a net loss of nutrients from the soil, and thus have severe consequences for the sustainability of the system in the long term.

**Author Contributions:** Conceptualization, F.J.S.-S.; data curation, M.T.-G.; formal analysis, L.R.y.A. and J.K.-V.; investigation, M.T.-G., O.O.Á.-R., L.R.y.A., J.K.-V. and F.J.S.-S.; methodology, M.T.-G., O.O.Á.-R. and F.J.S.-S.; project administration, F.J.S.-S.; supervision, F.J.S.-S.; visualization, M.T.-G., O.O.Á.-R., L.R.y.A., J.K.-V. and F.J.S.-S.; writing—original draft, M.T.-G., O.O.Á.-R., L.R.y.A., J.K.-V. and F.J.S.-S.; writing—review and editing, M.T.-G., O.O.Á.-R., L.R.y.A., J.K.-V. and F.J.S.-S. All authors have read and agreed to the published version of the manuscript.

**Funding:** Research was financed by the Mexican Secretary of Public Education (SEP) through the PROMEP program as well as the Priori program of the University of Yucatan and partially financed by IAEA (MEX/5/0266).

**Institutional Review Board Statement:** Not applicable.

**Informed Consent Statement:** Not applicable.

**Data Availability Statement:** For additional information, contact the author by correspondence.

**Conflicts of Interest:** The authors declare no conflict of interest.

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
