# Peer review of "Productivity and Nutritional Quality of Nitrogen-Fixing and Non-Fixing Shrub Species for Ruminant Production"

_agronomy, doi:10.3390/agronomy13041089_

Round 1

Reviewer 1 Report

Line

Comment - see highlighted text

Dates

There was a single reference to 2003 in the paper at line 286, but no other dates were mentioned. The climate figure needs to note which 2 calendar years are being described, and methods need to note the day, month and year of planting.

Genus vs. species names

There is only one species of each genus used in this study, and most commonly the genus name is used. On occasion, the authors switch to binomial nomenclature (e.g., L. leucocephala) for no particular reason, which is confusing. Unless there is a specific purpose (in the abstract or introduction, when the species (and cultivar, where applicable) needs to be specified, the authors should be consistent and use one or the other. It would seem simplest to use the genus names throughout.

14

Replace “in mixture” with “as a mixture”

15

Add “shrubs” to “non-leguminous”

31

Replace “events make” with “event makes”

57

Replace “relocated” with “relocate”

60

Replace “system, this” with “system. This”

67-69

References needed for this lists of characteristics.

76-77

This needs to be more specific. What treatments are being imposed? Why were they selected, and what do the authors expect to learn?

89

Replace “vegetation, dominant” with “vegetation. Dominant”

94

What are flat plastic bags? Do you mean the plastic bags with pleated bottoms? Include a brand and a more complete description.

97-98

Clarify “the seeds of all 3 species were sown directly in the soil-filled bags.” Do you mean that single seeds were grown into a seedling in each bag? This could be interpreted to mean that each bag contained a seed of each of the three seedlings.

128

Replace “pruning’s” with prunings” – this is plural but not possesive

132

Delete “up”

133

Replace “mesh” with “screen”

144

Replace “rate” with “rates”

152 & 154

Replace “Leucaena growing” with “the Leucaena component growing”

171

Change to “completely dry in Year 2”

204

Replace “and” with “but”

220

Replace “than Guazuma” with “than monoculture Guazuma

222

Replace “However, there” with “There”

236; 241, etc., and Fig. 2 X-axis

In the text and in Table 4, the authors are suddenly using Leucaena (G) and Leucaena (M) to represent the Leucaena-Guazuma and Leucaena-Moringa mixtures. In Tables 2 and 3, these notations represented the Leucaena component of those mixtures. Therefore, in Table 4 Leucaena (G) and Leucaena (M) should be replaced with Leucaena-Guazuma and Leucaena-Moringa to indicate the mixtures and not one component of the mixtures.

246

Replace “between” with “among”

258

Replace “increase” with “increased”

Fig. 2

In the legend, replace “Edib” with “Edible”

349

“mixtures” is the end of a sentence, so “[28]” needs to be the beginning of another sentence.

Table 5

In this table there are columns for polyphenols, total tannin and condensed tannins, but in the methods section there are only methods referenced for polyphenols and condensed tannins (line 137). What methodology and calculations are the basis for total tannins, and how do they differ from condensed tannins?

406

The singular of “species” is “species,” not “specie”

Author Response

We appreciate the comments made by the Reviewer. We attach a .PDF document with the responses to each of the comments made.

Reviewer 2 Report

This manuscript is in need for a thorough revision. At first an English language competent person should be consulted over the manuscript. Secondly, results presented should be checked again since they appear inconsistent in some cases. Table legends should be accurate in describing their contents. Discussion should not be hollow, but to comment solidly on the study. More comments are on the attached pdf file. 

Author Response

(The authors gave the same response as above.)
